# Correlated mechanochemical maps of *Arabidopsis thaliana* primary cell walls using atomic force microscope infrared spectroscopy

Natasha Bilkey[1] , Huiyong Li[2] , Nikolay Borodinov[3], Anton V. Ievlev[3],
Olga S. Ovchinnikova[3], Ram Dixit[1] and Marcus Foston[2]

[1]Department of Biology, Center for Engineering Mechanobiology, Washington University in St. Louis, St. Louis, Missouri 63130, USA; [2]Department of Energy, Environmental and Chemical Engineering, Center for Engineering Mechanobiology, Washington University in St. Louis, St. Louis, Missouri 63130, USA; [3]Center for Nanophase Materials Science, Oak Ridge National Laboratory, Oak Ridge, Tennessee 37831, USA

## Original Research Article

**Keywords:**
atomic force microscopy; infrared spectroscopy; mechanochemical properties; plant cell wall.

Na.B. and H.L. contributed equally to this work.

**Author for correspondence:**
M. Foston, E-mail: mfoston@wustl.edu

## Abstract

Spatial heterogeneity in composition and organisation of the primary cell wall affects the mechanics of cellular morphogenesis. However, directly correlating cell wall composition, organisation and mechanics has been challenging. To overcome this barrier, we applied atomic force microscopy coupled with infrared (AFM-IR) spectroscopy to generate spatially correlated maps of chemical and mechanical properties for paraformaldehyde-fixed, intact *Arabidopsis thaliana* epidermal cell walls. AFM-IR spectra were deconvoluted by non-negative matrix factorisation (NMF) into a linear combination of IR spectral factors representing sets of chemical groups comprising different cell wall components. This approach enables quantification of chemical composition from IR spectral signatures and visualisation of chemical heterogeneity at nanometer resolution. Cross-correlation analysis of the spatial distribution of NMFs and mechanical properties suggests that the carbohydrate composition of cell wall junctions correlates with increased local stiffness. Together, our work establishes new methodology to use AFM-IR for the mechanochemical analysis of intact plant primary cell walls.

## 1. Introduction

Cell walls are a defining feature of plant cells and play a critical role in protecting plants against environmental stress, providing structural support and determining the direction of growth. All plant cells have an extensible primary cell wall, which forms first and underlies cell growth and morphogenesis. Certain cell types, such as tracheids and fibre cells in the vasculature of the plant, develop a thick and non-extensible secondary cell wall during their differentiation. The primary cell wall consists of polysaccharides such as cellulose, hemicellulose and pectin, along with some proteins (Cosgrove, 2005; Lampugnani et al., 2018). Cells control the synthesis and deposition of individual cell wall components to modulate the composition of the cell wall. Spatial heterogeneity in cell wall composition is particularly important in defining the mechanical properties of the wall (Cosgrove, 2018; Grones et al., 2019; Majda et al., 2017; Phyo et al., 2017).

Cellulose is a linear polymer of $\beta$-1,4-linked glucose (Glc) that forms microfibrils which resist tensile stress and control the axis of turgor-driven cell expansion (Cosgrove, 2018). Cellulose microfibrils are believed to be mechanically coupled to each other at sites called 'biomechanical hotspots' where hemicelluloses, such as xyloglucan (XG), are thought to form tight linkages between microfibrils (Cosgrove, 2014; Park & Cosgrove, 2012). XG, the major hemicellulose in primary cell walls of *Arabidopsis thaliana* and other eudicots (Park & Cosgrove, 2015), consists of a $\beta$-1,4-Glc backbone substituted at regular positions with $\alpha$-D-xylose (Xyl). The Xyl monomers in XG are further substituted with $\beta$-D-galactose (Gal) and $\alpha$-L-fucose (Fuc). Another hemicellulose found in the primary cell walls of *A. thaliana* is glucuronoarabinoxylans, which consists of a $\beta$-1,4-xylose monomer backbone with sidechains of $\alpha$-D-arabinose (Ara) and/or $\alpha$-D-glucuronic acid (GlcA) monomers (Zablackis et al., 1995). The cellulose microfibrils and hemicellulose network is surrounded by a hydrophilic pectin matrix that aids in cell wall extensibility, strength and porosity (Atmodjo et al., 2013; Cosgrove, 2014; Mohnen, 2008; Willats et al., 2001). Pectin is a highly

branched polymer consisting of anionic sugars such as $\alpha$-D-galacturonic acid (GalA) and can undergo reversible methyl esterification to modulate cell wall elasticity (Höfte et al., 2012; Peaucelle et al., 2015). The main pectin in *A. thaliana* includes homogalacturonan (HGA), rhamnogalacturonan I (RG-I) and rhamnogalacturonan II (RG-II). Adjacent cells are adhered to each other via a pectin-rich middle lamella that is important for maintaining the structural integrity of plants (Bouton et al., 2002; Daher & Braybrook, 2015). Approximately 10% of the mass of the plant primary cell wall is comprised of enzymes and structural proteins. Some structural proteins, such as arabinogalactan proteins (AGPs) (Jamet et al., 2006), are glycosylated by oligosaccharides composed mainly of neutral sugars and a small number of GlcA monomers.

A variety of techniques have been developed to spatially visualise plant cell wall composition. For example, metabolic labelling with an alkynylated fucose analogue is used to fluorescently label the location of RG pectin in the cell wall (Anderson & Wallace, 2012; Anderson et al., 2012). Fluorescent dyes, such as propidium iodide and Pontamine fast scarlet 4SB, are used to stain and image the location of HGA pectin and cellulose, respectively (Anderson et al., 2010; Rounds et al., 2011; Thomas et al., 2013). Since these methods enable live imaging, they reveal real-time spatial locations and concentrations of specific wall components. However, these methods require labelling and typically focus on one cell wall epitope or component at a time.

Spectroscopy has been used as a label-free method to spatially analyse the cell wall composition of plant cells and tissues. For example, IR and Raman spectroscopy, which detect molecular vibrations via light absorption and inelastic scattering of photons, respectively, have enabled chemical imaging of plant cell walls and other biological materials (Baker et al., 2014; Gierlinger, 2014; 2017). Fourier-transform infrared spectroscopy (FTIR) and attenuated total reflection-FTIR microspectroscopy methods have been used to map cellulose- and lignin-rich regions in *Populus* wood cell walls (Canteri et al., 2019; Cuello et al., 2020; Gierlinger, 2017). Confocal Raman microspectroscopy has also been used to map cellulose- and lignin-rich regions in multiple plant types (Agarwal, 2006; Gierlinger, 2014; 2017; Gierlinger & Schwanninger, 2006; Prats Mateu et al., 2016; Schmidt et al., 2010). While these label-free techniques can simultaneously detect multiple cell wall components, they cannot offer insights into how these cell wall components correlate with cell wall mechanical properties (e.g., stiffness).

Atomic force microscopy (AFM) is a label-free method that can surpass the visible-light diffraction limit to visualise cell wall topography and mechanical properties at the nanoscale (Zhang et al., 2016). However, AFM does not directly provide chemical information, and cell wall composition can only be indirectly inferred from mechanical information and/or through changes in mechanical properties following treatments that degrade specific cell wall components or linkages (Marga et al., 2005; Zhang et al., 2016; 2017; 2019). AFM coupled with IR (AFM-IR) is an emerging technique to perform nanoscale multimodal imaging of physical, chemical and mechanical material features (Dazzi & Prater, 2017; Dazzi et al., 2005; Kurouski et al., 2020). This technique has been primarily used to generate correlated mechanochemical maps of polymer composites, films, fibres and blends (Centrone, 2015; Dazzi & Prater, 2017). AFM-IR uses a tunable IR laser focused on the sample at the AFM tip. When the wavelength of the IR laser matches the excitation wavelength of sample beneath the AFM tip, the sample absorbs the IR energy and undergoes thermal expansion which is detected by the AFM tip in contact with the sample surface. This can provide chemical images at selected IR wavelengths with nanoscale spatial resolution. Recently, AFM-IR has been used to image biological specimens such as triacylglycerol-containing vesicles from bacteria, yeast and microalga (Dazzi & Prater, 2017), extracellular vesicles from animal stem cells (Kim et al., 2018), bone (Gourion-Arsiquaud et al., 2014) and breast cancer cells (Clède et al., 2013; Dazzi & Prater, 2017). In plants, the application of AFM-IR has so far been limited to secondary cell walls (Farahi et al., 2017; Tetard et al., 2010; 2011; 2012; 2015).

In this work, we establish a new methodology for applying AFM-IR to simultaneously measure mechanical and chemical properties of the primary cell wall of *A. thaliana* epidermal cells. We provide a sample preparation technique suited for AFM-IR on *A. thaliana* stem sections, which eliminates chemical contamination. Since many plant cell wall components have overlapping spectral features and because IR spectral imaging creates large and complex datasets, we applied non-negative matrix factorisation (NMF) to reduce the dimensionality of the AFM-IR data into a more manageable number of components (Borodinov et al., 2019; Montcuquet et al., 2010). These non-negative matrix factors were used to identify the IR spectral features representing different cell wall components. By correlating the spatial distribution of chemical groups with nanomechanical properties, this study reveals how the spatial variation in cell wall composition relates to the spatial variation in cell wall mechanics.

## 2. Results and discussion

### 2.1. AFM-IR imaging of A. thaliana epidermal cell walls

The spatial resolution and signal quality of AFM-IR images depend greatly on sample thickness and roughness (Pereira et al., 2018). For example, in thick samples (>1 μm) of poor thermal conductivity, insufficient decay of thermal expansion with respect to scan rate can reduce spatial resolution and signal quality (Kurouski et al., 2020). Most AFM-IR studies have involved spun-cast polymer thin films rather than complex biological specimens. As a result, our study required special considerations for sample preparation.

Resin embedding is a commonly used technique to obtain thin sections with well-preserved cellular structures (Verhertbruggen et al., 2017). However, epoxy resin has strong IR absorption in the 1720–922 cm$^{-1}$ region, which would interfere with the IR signatures useful in our study (Coste et al., 2021; González et al., 2012; Pereira et al., 2018). Therefore, paraformaldehyde (PFA) tissue fixation and low melting temperature gelatin were used to embed apical portions of *A. thaliana* stems. A cryostat was then used to obtain cross sections from a frozen block of gelatin-embedded stem with controlled and uniform thickness (~10 μm) and surface roughness (<100 nm). This method resulted in sections that had intact epidermis, phloem and xylem cells, but softer tissues, like the pith and cortex, were less well preserved (Supplementary Figure S1A). AFM-IR imaging was conducted on the epidermal cell wall because the epidermis is composed of mostly primary cell walls (Cosgrove, 2018). Because sections were obtained perpendicular to the periclinal cell walls, the mechanical properties detected by AFM are unaffected by turgor pressure. PFA-fixed sections demonstrate minimal spectral difference from *A. thaliana* stem tissue that had undergone no sample preparation (Supplementary Figure S2). Therefore, the sample preparation method devised here was suitable for generating plant stem sections for AFM-IR with no detectable spectral contamination.

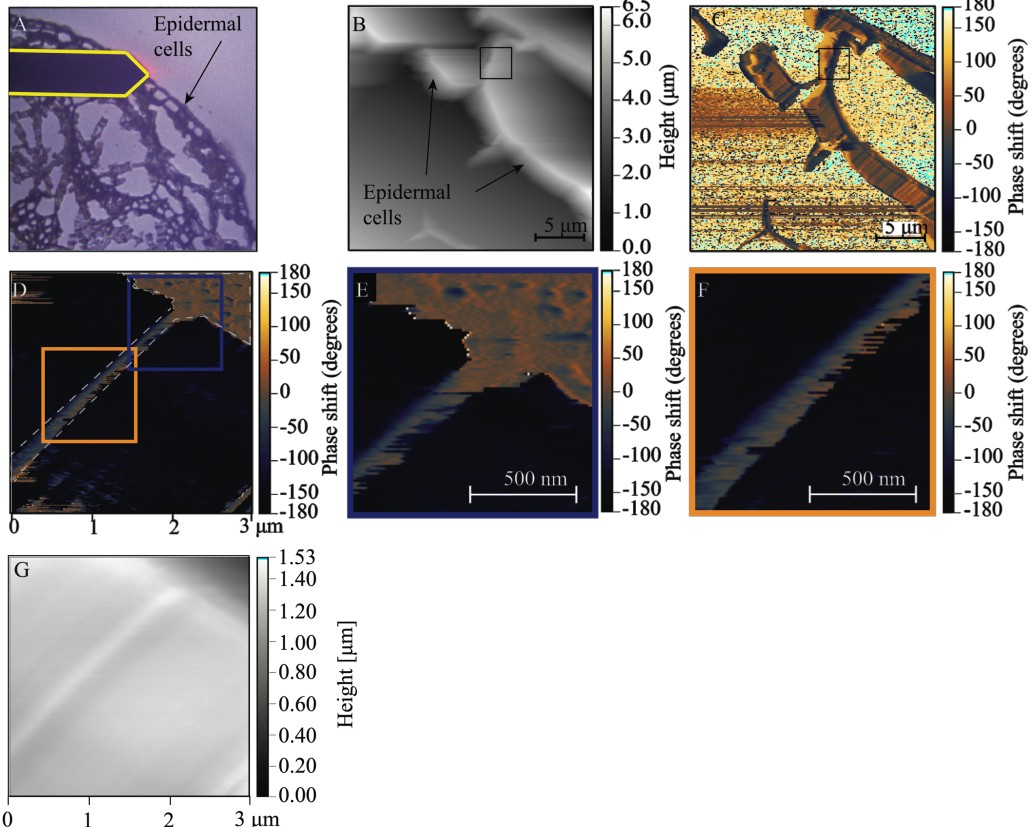

**Fig. 1.** AFM-IR imaging of an epidermal cell wall in a PFA-fixed *A. thaliana* stem sample. (a) Image of sample region from the AFM camera. AFM cantilever tip is outlined in yellow. (b,c) Contact resonance mode AFM height image and phase shift map of the large scan region ($25 \times 25$ $\mu m^2$) showing epidermal cells. Positive shifts in phase indicate increasing sample stiffness. Black box represents where AFM-IR measurements ($30 \times 30$ spectral array) were collected. Scale bar = 5 $\mu m$. (d) Phase shift map of the small scan region ($3 \times 3$ $\mu m^2$) where AFM-IR measurements were collected. White dashed line is outlining a region of increased stiffness at the junction between two epidermal cells. (e,f) Zoomed-in images of two regions of varying phase shift along cell–cell junction, indicated in (d). Scale bar = 500 nm. (g) Height map was obtained of a $3 \times 3$ $\mu m^2$ region where AFM-IR data were collected.

Topographical (Figure 1a,b) and phase (Figure 1c) images obtained via contact resonance mode AFM imaging were used to locate an epidermal cell wall. The topographical map of a $25 \times 25$ $\mu m^2$ region shows the wall between two adjoining epidermal cells and corresponds to the AFM-IR image collected (Figure 1b). Phase images report the difference in AFM cantilever drive and response oscillation that is sensitive to the surface stiffness and adhesion of the surface to the AFM tip (Ye & Zhao, 2010). Generally, softer samples result in a negative phase shift and stiffer samples result in a positive phase shift (Dong & Yu, 2003). Phase shift can be sensitive to cell wall composition and has been previously used to capture different cell wall ultrastructural features in young poplar stems (Farahi et al., 2017). Similarly, in the *A. thaliana* sample from this study, the phase map (Figure 1c) showed striations along the wall, presumably with regions of softer and stiffer material. The spatial variation in sample surface and subsurface properties indicated by phase shift can be attributed to spatial variation in cell wall mechanics and/or adhesion (Bidhendi & Geitmann, 2019), which are thought to be determined by wall composition and organisation (Bidhendi et al., 2019; Bidhendi & Geitmann, 2019; Bou Daher et al., 2018; Cosgrove, 2018; Peaucelle et al., 2011). For example, streaks in the phase shift maps indicate regions where the tip goes from an attractive (stiff) to a repulsive (soft) material (Figure 1d–f).

Chemical information on a shared epidermal cell wall was acquired with a $30 \times 30$ array (900 AFM-IR pixel points or spectra) over a $3 \times 3$ $\mu m^2$ region, collecting a matrix of spectra every

100 nm in the x- and y-dimension (black square in Figure 1b,c). Although the spatial resolution of our AFM-IR acquisition could not be determined, it nonetheless offered multimodal images to correlate cell wall chemical and mechanical properties. Within the scan region, images ($256 \times 256$ pixel map over a $3 \times 3$ $\mu m^2$ region) were collected for phase shift (Figure 1d), height (Figure 1g), phase-locked loop (PLL) frequency (Supplementary Figure S1B) and deflection (Supplementary Figures S1C, E). Continuous adjustment in PLL frequency is necessary to maintain the resonance-enhanced condition throughout the scan. Hence, PLL monitors changes in contact resonance frequency which occurs due to changes in sample surface stiffness and adhesion (Stan & Solares, 2014). Deflection is a direct measure of cantilever-surface interaction force and is used to produce a three-dimensional image of the sample surface (Vadillo-Rodríguez et al., 2004). Based on Figure 1, the stiff portion of the wall, indicated by high positive phase shift, is outlined by a white dashed line (Figure 1d). This region appears slightly raised in the height map (Figure 1g), with more distinct three-dimensional structure in the deflection (Supplementary Figure S1C) and lateral deflection (Supplementary Figure S1D) maps. Based on the orientation of the sample, this region was determined to be the junction between the anticlinal cell walls of two epidermal cells. Figure 1e and f shows enlarged phase shift images of the cell–cell junction, suggesting a stiffer boundary between the cells and softer regions on either side. This stiff cell–cell junction likely represents the middle lamella, a pectin-

rich region that acts as a 'glue' between shared cell walls (Daher & Braybrook, 2015; Zamil & Geitmann, 2017). Previous work by Zamil et al. (2014) using a micro-stretching technique also found that the middle lamella between epidermal cells is significantly stronger than the rest of the cell wall.

## 2.2. Determining the spatial variation in primary cell wall composition by AFM-IR

In our experiments, the available lasers for AFM-IR spectral acquisition were limited to a range of 922–1170, 1301–1410 and 1504–1720 cm$^{-1}$. All FTIR and AFM-IR spectra in this study were pre-processed applying spectral smoothing, baseline correction and normalisation approaches (Gautam et al., 2015) (see Supporting Information and Supplementary Figure S3). To ensure that spectra produced by AFM-IR are similar to those obtained by traditional FTIR, principal component analysis (PCA) was used to compare FTIR and AFM-IR spectra of carbohydrate standards obtained from plant primary cell wall. The results show that the FTIR and AFM-IR spectra of the carbohydrate standards are statistically indistinguishable (Supplementary Figure S2). Therefore, the AFM-IR spectra accurately reflect cell wall IR response and chemistry.

The IR spectra of the primary cell wall are complex because the different polymeric components each have intricate chemical functionality. FTIR spectra of cell wall standards were used to establish IR band assignments to specific primary cell wall components. Commercially available XG, RG-I, cellulose, xylan and polygalacturonic acid (PGA) samples extracted from various plant sources were used as representative examples of cellulose, pectin and hemicellulose. Lignin was not included as a cell wall standard since it is not abundant in the epidermal cell walls of *A. thaliana* apical stems (Zhu et al., 2015). Further, imaging of lignin autofluorescence (Decou et al., 2017) demonstrated that there is essentially no lignin in the epidermis (Supplementary Figure S4). FTIR spectra of each standard are shown in Figure 2a, and major IR bands observed for each cell wall standard are detailed in Supplementary Table S1. To further facilitate interpretation of AFM-IR spectral data, FTIR spectra of *A. thaliana* primary cell wall reference fractions were obtained. Oxalate, 1 M KOH and 4 M KOH extracts were isolated from alcohol-insoluble residues generated from *A. thaliana* seedling by sequential extraction (Supplementary Figure S5). Previously, Peralta et al. performed sequential extractions followed by glycome profiling on the apical section of an *A. thaliana* stem (Pattathil et al., 2012; Peralta et al., 2017). It was found that the oxalate extract was enriched in RG-I and AG epitopes (Peralta et al., 2017). Similarly, in this study, the oxalate extract isolated from *A. thaliana* seedlings was found to consist mainly of Gal (39%), Ara (25%) and GalA (20%) residues (Supplementary Table S2). In addition to carbohydrates, this extract also contained a significant amount of protein (see Supplementary Table S3). Therefore, the oxalate extract, in this study, likely contains pectins, AGPs and other cell wall proteins. Peralta et al. (2017) found that the subsequent carbonate and 1 M NaOH extracts were enriched in epitopes associated with xylans, HGA, RG-I and AG. In this study, the polysaccharides in the 1 M KOH extract were composed mainly of Xyl (35%), Ara (21%), Gal (20%), GalA (8%) and Glc (6%) residues (Supplementary Table S2). The 1 M KOH extract also contains a significant amount of protein (Supplementary Table S3). Therefore, the1 M KOH extract, in this study, likely contains xylans, pectins and cell wall proteins. Lastly, Peralta et al. (2017) reported that a 4 M NaOH extract was enriched in epitopes associated with non-fucosylated XGs, fucosylated XGs and xylan. In this study, the

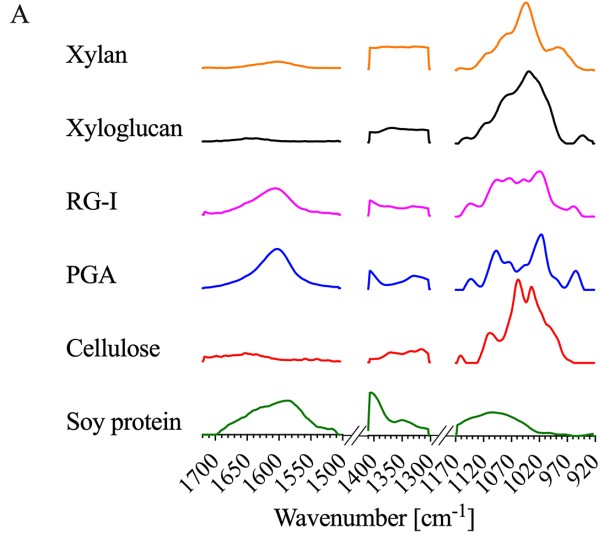

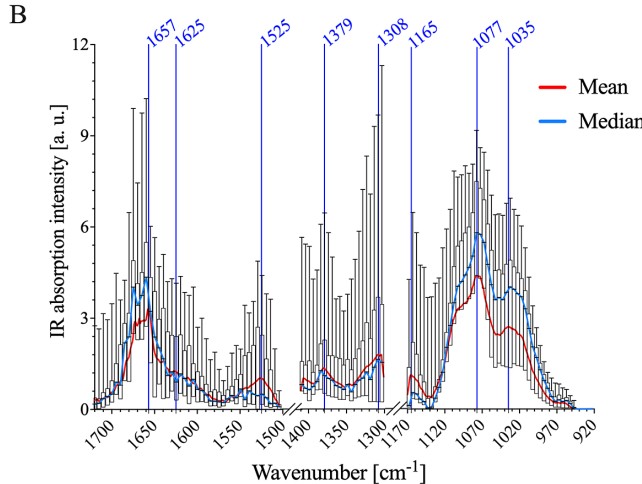

**Fig. 2.** AFM-IR data of PFA-fixed *A. thaliana* stem epidermal cell wall show similar absorption patterns to FTIR spectra of cell wall standards. (a) Normalised FTIR spectra of cellulose, soy protein, PGA, RG-I, xylan and xyloglucan. FTIR absorption between 1,300–1,171 and 1,503–1,411 cm$^{-1}$ was removed for comparison with AFM-IR data. (b) A box plot of the AFM-IR spectra collected on the epidermal cell wall sample surface by binning 30×30 pixels into 15×15 pixels and by binning spectral wavenumbers 8×. The red and blue lines represent the mean and median AFM-IR spectra, respectively. There was no AFM-IR absorption between 1503–1411 and 1300–1171 cm$^{-1}$.

polysaccharides in the 4 M KOH extract were composed mainly of GalA (25%), Xyl (24%), Ara (14%), Gal (13%) and Glc (10%) residues (see Supplementary Table S2). In addition, the 4 M KOH extract lacked significant amounts of protein (Supplementary Table S3). This dataset suggests that the 4 M KOH extract, in this study, likely consists of XGs, pectins and xylans.

The cellulose, xylan and XG standards exhibit strong IR absorption between ~975 and 1170 cm$^{-1}$ (Figure 2a). As shown in Table 1, these IR bands are related to: (a) sugar ring ether O-C-O, (b) primary and secondary alcohol C-O and (c) glycosidic ether O-C-O stretching commonly observed in the neutral residues and polysaccharide linkages of cell wall carbohydrates. The oxalate, 1 M KOH and 4 M KOH reference extracts, as well as the mean AFM-IR spectrum of the *A. thaliana* epidermal cell wall, also show strong IR absorption in the ~975–1170 cm$^{-1}$ region (Supplementary Figure S5 and Figure 2b), suggesting the presence of carbohydrates. The mean *A. thaliana* AFM-IR spectrum contains prominent bands

**Table 1.** IR assignments of the average Arabidopsis AFM-IR spectrum compared to commercial standards.

| Wavenumber range (cm$^{-1}$) | Wavenumber maxima (cm$^{-1}$) | Band assignments | Commercial standards |
| --- | --- | --- | --- |
| 1706–1631 | 1657 | Amide I (C=O stretch weakly coupled with C-N stretch and N-H bending) | Soy protein |
| 1631–1598 | 1625 | Adsorbed water O–H bending | Cellulose, xylan, xyloglucan |
| 1631–1598 | 1625 | COO- antisymmetric stretching | PGA, RG-I |
| 1631–1598 | 1625 | Adsorbed water O–H bending | PGA, RG-I |
| 1567–1504 | 1525 | Amide II (C-N stretch strongly coupled with N-H bending) | Soy protein |
| 1394–1351 | 1379 | Alcohol O-H bending | Cellulose, xylan, xyloglucan, PGA, RG-I |
| 1394–1351 | 1379 | Aliphatic C-H bending | Cellulose, xylan, xyloglucan, PGA, RG-I |
| 1394–1351 | 1379 | COO- symmetric stretching | PGA, RG-I |
| 1342–1301 | 1308 | C-H bending Alcohol O-H bending | Cellulose, xylan, xyloglucan, PGA, RG-I |
| 1342–1301 | 1308 | Amide III (N-H in plane bending coupled with C-N stretching) | Soy protein |
| 1170–1145 | 1165 | Glycosidic O-C-O vibrations | Cellulose, xylan, xyloglucan, PGA, RG-I |
| 1137–1045 | 1077 | Sugar ring ether O-C-O stretching | Cellulose, xylan, xyloglucan, PGA, RG-I |
| 1137–1045 | 1077 | C-O stretching (secondary alcohol) | Cellulose, xylan, xyloglucan, PGA, RG-I |
| 1045–944 | 1035 | C-O and C-C stretching (primary/secondary alcohol) | Cellulose, xylan, xyloglucan, PGA, RG-I |

with peak maxima at: (a) 1310 cm$^{-1}$ assigned to alcohol O-H bending and/or the amide III band; (b) 1380 cm$^{-1}$ assigned to alcohol O-H bending, aliphatic C-H bending and/or carboxylic acid symmetric stretching; (c) 1525 cm$^{-1}$ assigned to the amine II band and (d) 1670–1660 cm$^{-1}$ assigned to adsorbed water O-H bending, carboxylic acid antisymmetric stretching and/or the amine I band. Prominent IR bands at 1310, 1380, 1525 and 1657 cm$^{-1}$ are also observed for the soy protein standard (Figure 2a), indicating the presence of cell wall proteins in the *A. thaliana* epidermal cell wall spectrum. While the oxalate, 1 M KOH and 4 M KOH reference extracts also absorb at 1525 cm$^{-1}$ (Supplementary Figure S5), the relative absorption peak intensity at 1525–1075 cm$^{-1}$ for the 4 M KOH reference extract is much lower than for oxalate and 1 M KOH reference extracts. The lower intensity in this region suggests that the 4 M KOH reference extract contains significantly less protein. Pectin cell wall standards (i.e., PGA and RG-I) show strong IR bands at ~1407 and 1604 cm$^{-1}$, which are assigned to carboxylic acid symmetric and antisymmetric stretching (Figure 2a). Although IR absorption at ~1407 and 1604 cm$^{-1}$ is observed in the mean *A. thaliana* AFM-IR spectrum (Figure 2b) and in the oxalate, 1 M KOH and 4 M KOH reference extracts (Supplementary Figure S5), these bands overlap with the IR absorption of proteins, which makes resolving the presence of acidic sugars on pectin based on carboxylic acid functionality difficult. PCA analysis further specifies the extent of chemical similarity between cell wall reference fractions, cell wall standards and the *A. thaliana* epidermal cell wall regions (Supplementary Figure S6).

The variance of IR absorption intensity across measured wavenumbers is substantial (Figure 2b), suggesting that there is chemical complexity and heterogeneity as a function of AFM-IR pixel location. Single wavenumber maps (Figure 3) were generated to depict the spatial variation in IR absorption of specific functional groups associated with different cell wall standards (Table 1): neutral sugars in carbohydrates (C-O; 1075 cm$^{-1}$); amide II in proteins (C-N; 1525 cm$^{-1}$) and acidic sugar in carbohydrates, amide I in protein and/or adsorbed water (1660 cm$^{-1}$). While both 1075 and 1660 cm$^{-1}$ appear to have high absorption intensity across the majority of the AFM-IR scan region, there are clear differences in the level of absorption at these wavenumbers and their spatial distribution. For example, the area that shows the strongest absorption at 1075 cm$^{-1}$ shows weakest absorption at 1525 and 1660 cm$^{-1}$. Similarly, the region towards the bottom of the map that shows strong absorption at 1660 cm$^{-1}$ also shows weak absorption at 1075 and 1525 cm$^{-1}$. By contrast, in the top third of the map (region between 2.5 and 3.0 μm in the y-dimension), there is relatively equal spatial distribution of all three wavenumbers. Thus, single wavenumber maps can distinguish between the spatial distribution and concentration of spectrally resolved functional groups, such as C-O and C=O. However, these functional groups are found in multiple cell wall components. Therefore, single wavenumber maps are insufficient to fully capture the spatial and chemical complexity of cell wall components.

## 2.3. Deconvoluting AFM-IR spectra for mechanochemical correlations

To more accurately delineate local concentrations of specific cell wall functional groups and components, NMF spectral deconvolution was applied (Borodinov et al., 2019; Kulkarni et al., 2018; Labbe et al., 2005; Lin et al., 2018; Montcuquet et al., 2010; Zhang et al., 2018). NMF of the AFM-IR dataset (i.e., input matrix) requires a subjective, user-specified number of factors. The cophenetic corre-

lation coefficient and residual sum of squares for a given solution represent the extent to which the residual data (i.e., the data from the input matrix that was not recreated by linear combinations of the factors) can be explained by best-fit linear combinations of solution factors. For example, Brunet et al. (2004) proposed that the cophenetic correlation coefficients demonstrate the stability of NMF solutions or how accurately the NMF solution describes the dataset. Hutchins et al. (2008) proposed that the proper NMF solution occurs when the residual sum of square, or the variation between the target matrix and the NMF solution, is minimised. However, these measures alone cannot solely inform which is the 'best' solution or number of factors. In this study, NMF solutions were generated with the number of factors ranging from two to six. Distinctive factor spectra were generated up to the three-factor solution, but following that point, increasing factor number resulted simply in factor splitting. Furthermore, the cophenetic correlation coefficient and residual sum of squares values were relatively low for the three-factor solution (Supplementary Figure S7). Therefore, the three-factor NMF solution was used to analyse the *A. thaliana* AFM-IR dataset.

Key IR absorption bands in the NMF spectra were analysed and used to attribute the origin of NMFs (Figure 4a and Table 2). Factor 1 spectrum exhibited strong absorption for sugar ring ether O-C-O and primary alcohol C-O stretching at IR bands between 975 and 1091 cm$^{-1}$, which are absorption signatures associated with cell wall carbohydrates (Figure 4a). Absorption in this region was also seen in the FTIR spectra of carbohydrate standards and reference extracts. Notably, similar to cellulose, XG and xylan, the Factor 1 spectrum did not show strong absorption in regions related to carboxylic acids (1407 and 1604 cm$^{-1}$). Factor 1 thus primarily represents a set of chemical groups that comprise neutral sugars on primary cell wall carbohydrate components, although there are likely spectral elements attributed to cellulose, XG and xylan in Factor 2. The Factor 2 spectrum contained IR absorption bands between (a) 1063 and 1170 cm$^{-1}$ with peak maxima at 1110 and 1165 cm$^{-1}$ that are attributed to glycosidic ether O-C-O and/or secondary alcohol C-O stretching and (b) 1301 and 1346 cm$^{-1}$ with a peak maximum at 1310 cm$^{-1}$ that is attributed to alcohol O-H bending and/or the amide III band. These two IR absorption band regions are found in the FTIR spectra of both carbohydrate and protein standards. Factor 2 thus represents specific moieties on primary cell wall carbohydrate components, mainly glycosidic linkages, and N-H groups of amides on proteins. The Factor 3 spectrum contained IR absorption bands between (a) 1324 and 1410 cm$^{-1}$ with peak maxima at 1380 and 1403 cm$^{-1}$ that are attributed to alcohol O-H bending, aliphatic C-H bending and/or carboxylic acid symmetric stretching, (b) 1504 and 1570 cm$^{-1}$ with a peak maximum at 1525 cm$^{-1}$ that is attributed to the amide II band and (c) 1596 and 1696 cm$^{-1}$ with peak maxima 1610 and 1660 cm$^{-1}$ that are attributed to carboxylic acid antisymmetric stretching, adsorbed water O–H bending and/or the amide I band. Factor 3 thus represents the acidic sugars on primary cell wall carbohydrate components, mainly carbonyls, as well as C-N and C=O groups of amides on proteins. The FTIR of RG-I and PGA suggest that the IR band for carboxylic acid antisymmetric stretching is centred closer to 1600 cm$^{-1}$. Accordingly, a major IR absorption feature of the spectra of Factor 3 located more towards 1660 cm$^{-1}$ is likely more characteristic of the presence of a protein amide I band or protein-adsorbed water. Factor 1 loading was mapped in Figure 4b. As expected for the plant cell wall, neutral sugars represented by Factor 1 are present throughout most of the AFM-IR scan region. The NMF loading maps for Factors 2 and 3 are shown in 6 × 6 pixel

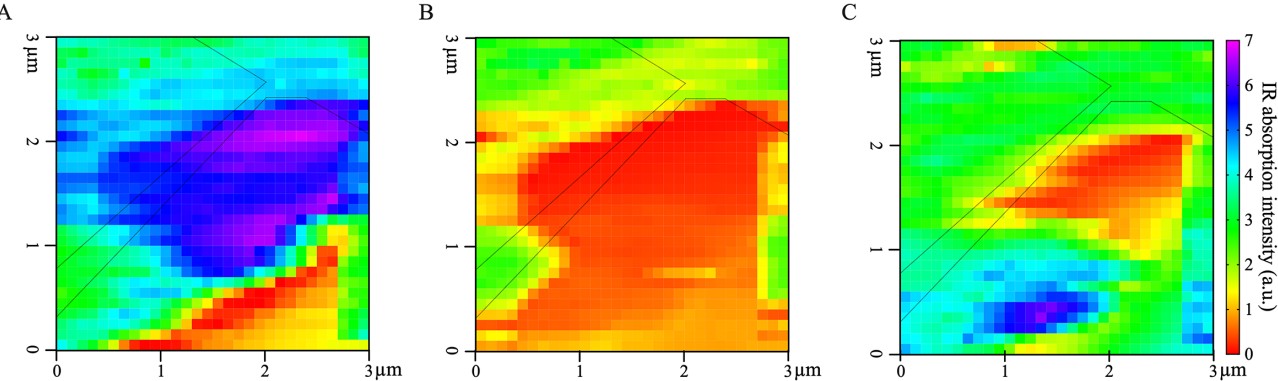

**Fig. 3.** IR wavenumber maps at (a) 1,075 cm$^{-1}$, (b) 1,525 cm$^{-1}$ and (c) 1,660 cm$^{-1}$. The black dashed outline represents the cell–cell junction highlighted in Figure 1d. a.u. = arbitrary units.

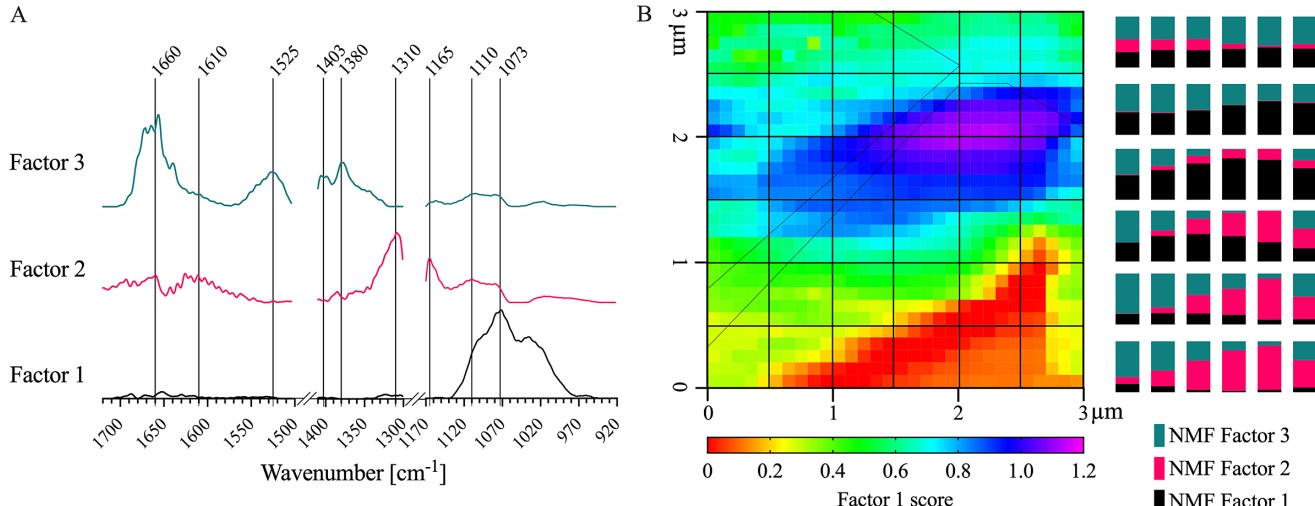

**Fig. 4.** NMF factor spectra represent carbohydrate-rich and protein-like regions in the epidermal cell wall. (a) NMF factor spectra plot for the 3-factor NMF solution. Peaks in each NMF factor spectrum are labelled with vertical lines. There was no AFM-IR absorption between 1,503–1,411 cm$^{-1}$ and 1,300–1,171 cm$^{-1}$. (b) Distribution map of Factor 1 representing its concentration (absolute value) in each pixel, with stacked bar plots displaying the ratio of NMF scores of Factors 1, 2 and 3 binned across 6×6 pixels (one box of the grid). Black dashed outline represents the cell–cell junction highlighted in Figure 1d.

squares, in agreement with the cell wall being a composite material (Supplementary Figure S8) (Burton et al., 2010).

The left and top edges of the *A. thaliana* AFM-IR image are enhanced in the abundance of Factor 3 spectral features, whereas the bottom-right corner of the *A. thaliana* AFM-IR is enhanced in the abundance of Factor 2 spectral features. To better visualise these differences, we generated a map of the difference between Factor 1 score and the sum of Factors 2 and 3 scores (Figure 5a). Although an operation of this type with NMF loading values is ill-defined, it can be used to visualise which locations are enhanced in Factor 1 versus Factors 2 and 3 or vice versa. Positive values indicate locations where the IR spectra are enhanced in Factor 1, while negative values indicate locations with IR spectra are enhanced in Factors 2 and 3. In this map, Factor 1 dominates an oval-shaped region along the cell–cell junction (cyan and blue pixels), suggesting a high relative abundance of neutral sugars in this region. This map suggests that cellulose, XG and xylan are present at higher concentrations than proteins and pectins within this region. By contrast, proteins and pectins represented by Factors 2 and 3 together appear to dominate the bottom third of the AFM-IR image (yellow and red pixels). Plants contain various cell wall proteins that make up 5–10% of the cell wall (Albenne et al., 2013; Amos & Mohnen, 2019; Daher & Braybrook, 2015; Keller, 1993). Localised deposition of

cell wall proteins was previously demonstrated by Smallwood et al. (1995), who performed immunohistochemical staining of extensin and showed it localises as puncta in the root epidermal cell wall. Without the need for immunostaining, our results further support that cell wall proteins may be preferentially deposited in certain regions of the cell wall. The remaining light-green coloured pixels in the map depict an equal mixture of Factor 1 and Factors 2 + 3, indicating that in some regions of the cell wall the concentration of cell wall components is more balanced. These results show that NMF is a useful statistical tool to deconvolute AFM-IR spectral datasets to identify the major chemical components of cell walls and to distinguish their spatial organisation and concentration.

### 2.4. Correlating cell wall composition and stiffness

An advantage of AFM-IR is its use in co-registering chemical and nanomechanical information. Cross-correlation coefficients can be used to evaluate the relationship between local mechanical properties, measured by cantilever phase shift, and chemical fingerprints (Borodinov et al., 2019). In this study, a positive cross-correlation value at a given pixel could result from either an increase in both the relative Factor 1 score and phase shift or an increase in relative combined Factors 2 and 3 scores and decrease in phase shift.

**Table 2.** IR assignments of NMF factors.

| Factor | Wavenumber range (cm$^{-1}$) | Wavenumber maxima (cm$^{-1}$) | Band assignments |
| --- | --- | --- | --- |
| 3 | 1696–1596 | 1660, 1610 | Amide I (C=O stretch weakly coupled with C-N stretch and N-H bending) |
| 3 | 1696–1596 | 1660, 1610 | Carboxylic acid O=C-O stretching |
| 3 | 1696–1596 | 1660, 1610 | Adsorbed water O–H bending |
| 2 | 1648–1543 | 1610 | Carboxylic acid O=C-O stretching |
| 2 | 1648–1543 | 1610 | Adsorbed water O–H bending |
| 3 | 1571–1504 | 1525 | Amide II (C-N stretch strongly coupled with N-H bending) |
| 3 | 1410–1324 | 1403, 1380 | Alcohol O-H bending |
| 3 | 1410–1324 | 1403, 1380 | Carboxylic acid O-H bending |
| 3 | 1410–1324 | 1403, 1380 | Aliphatic C-H bending |
| 2 | 1346–1301 | 1310 | Alcohol O-H bending |
| 2 | 1346–1301 | 1310 | Amide III (N-H in plane bending coupled with C-N stretching) |
| 2 | 1346–1301 | 1310 | Carboxylic acid O=C-O stretching |
| 2 | 1170–1063 | 1165, 1110 | Glycosidic ether O-C-O stretching |
| 2 | 1170–1063 | 1165, 1110 | Secondary alcohol C-O stretching |
| 1 | 1091–975 | 1073 | Sugar ring ether O-C-O stretching |
| 1 | 1091–975 | 1073 | Primary alcohol C-O stretching |

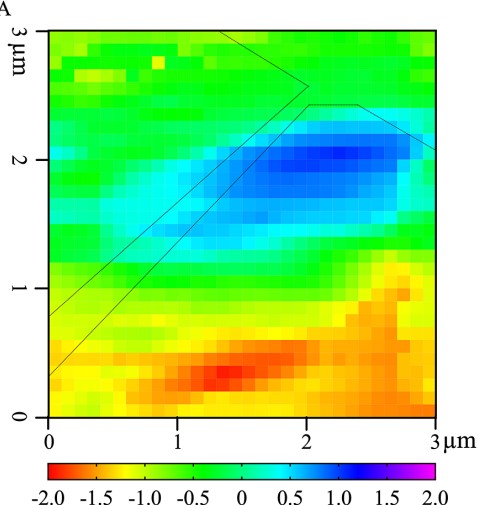

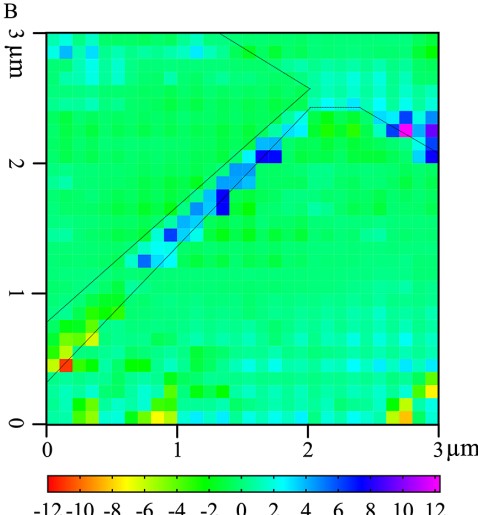

**Fig. 5.** Higher relative presence of NMF Factor 1 positively correlates with cell wall stiffness. (a) NMF distribution difference between Factor 1 score and sum of Factors 2 and 3 scores. (b). Correlation between phase shift and the NMF distribution difference between Factor 1 score and sum of Factors 2 and 3 scores as seen in Figure 4a for each pixel. Black dashed outline represents cell–cell junction highlighted in Figure 1d.

Conversely, a negative cross-correlation value at a given pixel could result from either an increase in relative Factor 1 score but decrease in phase shift or an increase in both the relative combined Factors 2 and 3 scores and phase shift. The former result would suggest that cell wall components described by Factor 1 correlate with cell wall stiffening while the cell wall components described by Factors 2 and 3 correlate with cell wall softening, while the latter result would suggest the reverse. The magnitude of the cross-correlation value is indicative of the strength of correlation.

The epidermal cell–cell junction displayed a distinguishable topological feature (Figure 1g) as well as appeared stiffer than the rest of the scanned region, as indicated by positive phase shift (Figure 1d). Cross-correlation analysis showed that higher relative abundance of neutral sugars represented by Factor 1 correlates with cell wall stiffening, specifically along the cell–cell junction (blue pixels; Figure 5b). It is possible that around the cell–cell junction, a higher concentration of crystalline cellulose is driving wall stiffening (Rongpipi et al., 2019). Combined Factors 2 and 3 scores correlate with cell wall stiffening in a small region along the

cell–cell junction (red and yellow pixels; Figure 5b). NMF factor score and phase shift in regions adjacent to the cell–cell junction are not strongly correlated, suggesting that softening of the cell is not solely due to the concentration of a particular cell wall component or, at least, the loading of a Factor 1 or Factors 2 + 3.

## 3. Conclusions

This study establishes new methodologies for sample preparation, data acquisition and multivariate statistical analyses for AFM-IR on *A. thaliana* primary cell walls. A method using gelatin embedding and cryostat sectioning of PFA-fixed *A. thaliana* stems was effective at eliminating spectral contamination and obtaining a section suitable for AFM-IR. Through adaptation of this sample preparation and sectioning method, AFM-IR may be applied to other genotypes, plant species and plant tissues to assess local cell wall mechanochemical properties. This work also demonstrated a novel application of NMF to analyse AFM-IR datasets. NMF deconvolution of AFM-IR spectra across the epidermal cell wall detected factors representing chemical functionality of neutral sugars of cell wall carbohydrates, acidic sugars of cell wall carbohydrates, and amino and carbonyl groups of cell wall proteins. Ultimately, cross-correlation analysis of the spatial distribution of NMFs and mechanical properties showed that the neutral sugar containing carbohydrate composition of cell wall junctions between neighbouring *A. thaliana* epidermal cells correlates with increased local stiffness. Future improvement of AFM-IR spatial resolution and spectral range may reveal more subtle patterns in component organisation that are important for defining local mechanical properties of the cell wall. With these advances in methodologies for sample preparation, data acquisition and multivariate statistical analyses, AFM-IR can be used to explore and understand phenotypic alterations to specific plant tissues in nanoscale resolution caused by environmental stresses or genetic mutations.

## 4. Materials and methods

### 4.1. Sample preparation

*Arabidopsis thaliana* ecotype Columbia-0 (Col-0) plants were grown under continuous light conditions ($120\ \mu Em^{-2}s^{-1}$) for 6 weeks. Apical regions of inflorescence stems were sectioned and fixed overnight with 3% PFA solution in phosphate-buffered saline (1.4 M NaCl, 1 mM KCl, 100 mM $Na_2PO_4$, 18 mM $KH_2PO_4$ pH 7.4). Fixed stem pieces were placed in a cryomold box containing room temperature 10% low-melting deep-sea cold fish gelatin (Cat. No. G7041, Sigma, St. Louis, USA) in deionised water. Stems were flash frozen in liquid nitrogen while held upright. Frozen gelatin blocks were sectioned using a cryostat (CM1850, Leica Microsystems) at 10 μm thickness using a Leica 819 microtome blade (Leica Biosystems Nussloch GmbH, Germany), washed three times with deionised water and then adhered to a zinc sulphide block and dried overnight in the fume hood. Sections were stored at room temperature until imaging.

### 4.2. AFM-IR measurement

IR spectral measurements were collected using a Nano-IR2-s instrument in contact resonance mode located at the Center for Nanophase Materials Sciences at Oak Ridge National Laboratory. Analysis Studio software (Bruker Corporation, Billerica, MA, USA)

monitored IR absorbance at wavenumber regions of 922–1170, 1301–1410 and 1504–1720 cm$^{-1}$. Soft IR probes with a nominal spring constant of 0.04 N/m were used. The PLL frequency was tuned to ~60 kHz, and the laser spot was optimised to the AFM tip with the largest IR absorption at 1030 cm$^{-1}$. Scans for topography, PLL frequency, phase shift, lateral deflection and deflection were first collected over a large $25 \times 25$ μm$^2$ region on epidermal cells at a scan rate of 0.5 Hz with a resolution of $128 \times 128$ pixel points and 8 co-averages in contact resonance mode. Within this region, the same scan types were then collected over a small $3 \times 3$ μm$^2$ region at a scan rate of 0.2 Hz with a resolution of $256 \times 256$ pixel points and 8 co-averages. A $30 \times 30$ IR array was collected in the small region with four co-averages and a spectral resolution of 1 cm$^{-1}$.

### 4.3. NMF and cross-correlation with phase shift

NMF analysis was conducted using factors varying in number from two to six to determine the optimal number of NMF factors, in part, based on cophenetic correlation coefficients and residual sum of squares. Once the optimised the number of NMF factors was determined, the NMF factor spectra (loading) and factor distribution coefficients (score) were exported. The solved NMF factor spectra were compared to the FTIR spectra of the standards using PCA. Then, NMF factor distribution maps were generated to visualise the factor spatial distribution. The phase shift map (originally 256 × 256 pixel points) collected on the $3 \times 3$ μm$^2$ region was reduced, by average data binning, to 30 × 30 pixel points, matching the NMF factor distribution map. Cross-correlation analysis between phase shift and the difference in NMF factor scores between Factor 1 and sum of Factors 2 and 3 was then performed similar to the protocol used in Borodinov et al. (2019).

### Acknowledgements

We thank Kirk Cyzmmek at the Donald Danforth Plant Science Center and Laurene Tetard from the Nano Science Technology Center at the University of Florida for their advice and guidance during this project. We would also like to thank Dr. Brent Williams in the Department of Energy, Environmental and Chemical Engineering at Washington University in St. Louis for his help with NMF analysis. We thank Sanja Sviben, Greg Strout and James Fitzpatrick from the Washington University Center for Cellular Imaging (WUCCI) for help with developing the sample preparation method which was supported by the Washington University School of Medicine, The Children's Discovery Institute of Washington University and St. Louis Children's Hospital (CDI-CORE-2015-505 and CDI-CORE-2019-813) and the Foundation for Barnes-Jewish Hospital (3770 and 4642). AFM-IR measurements were conducted at the Center for Nanophase Materials Sciences, which is a US DOE Office of Science User Facility supported under Contract (DE-AC05-00OR22725).

**Financial support.** This study was supported by the Center for Engineering Mechanobiology (CEMB), an NSF Science and Technology Center, under grant agreement CMMI: 15-48571. Na.B. was supported by a William H. Danforth Plant Sciences Fellowship.

**Conflicts of interest.** The authors declare no conflicts of interest.

**Authorship contributions.** Na.B. and H.L. carried out acquisition of AFM-IR datasets, developed python code, conducted data analysis and interpreted results. Ni.B. developed Python code for NMF analysis and assisted in application of NMF. A.V.I. and O.S.O. supervised data acquisition at Oak Ridge National Laboratory. R.D. and M.F. supervised the project. The manuscript was written through contributions of all authors. All authors have given approval to the final version of the manuscript.

**Data availability statement.** The data and coding supporting the findings of this study are available from the corresponding authors upon request.

**Supplementary Materials.** To view supplementary material for this article, please visit http://doi.org/10.1017/qpb.2022.20.

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
