## [Reviewer Report]

Dear Quantitative Plant Biology Editors:

We would like to submit our manuscript entitled, “Correlated mechanochemical maps of Arabidopsis thaliana primary cell walls using atomic force microscope-infrared spectroscopy” to be considered for publication as a research article in Quantitative Plant Biology.

Mechanical properties of the primary cell wall play a critical role in defining the growth axis and morphogenesis of plant cells. Cell wall mechanical properties and function are determined by the composition and organization of cell wall components. However, defining the relationship between cell wall composition and mechanics has been challenging due to the difficultly in simultaneously imaging both properties. Here, we use atomic force microscopy coupled with infrared-spectroscopy (AFM-IR) to generate co-localized, high-resolution chemical and mechanical maps of epidermal cell walls of a wild-type Arabidopsis thaliana stem section. This work necessitated a sample preparation method using commonly available reagents and materials aimed to minimize contamination of the cell wall IR spectrum. We developed a multivariate statistical analysis pipeline using non-negative matrix factorization (NMF) that visualizes cell wall chemical heterogeneity, assigns chemical origin to IR spectra, and resolves the spatial distribution of these chemical components at high resolution. NMF deconvolution of AFM-IR data revealed a linear combination of neutral sugars, acidic sugars, and amino group, and we found that localized concentration of neutral sugar-containing wall components stiffens the cell wall. 

Our findings demonstrate the combined power of AFM-IR and multivariate statistical analyses to elucidate how the organization of plant cell wall components determines local cell wall mechanics at the nanoscale. This application of AFM-IR lays the groundwork for future work to probe the nanoscale-level relationship between cell wall structure, chemistry, and mechanics for improving our understanding of plant growth and development. We believe that the methodology developed in this study will have broad and technical interest to the readers of Quantitative Plant Biology.

Reviewers we can suggest based on their acknowledged expertise in this area are: Notburga Gierlinger, Ph.D. (burgi.gierlinger@boku.ac.at); Janina Kneipp, Ph.D. (janina.kneipp@chemie.hu-berlin.de); Patrick Chaimbault, Ph.D. (patrick.chaimbault@univ-lorraine.fr); Sonia Contera, Ph.D. (sonia.antoranzcontera@physics.ox.ac.uk); and Joseph A. Turner, Ph.D. (jaturner@unl.edu).

Sincerely,

Ram Dixit

---

## [Reviewer Report]

*Comments to Author*: This paper successfully combines AFM with infrared spectroscopy to demonstrate that simultaneous sub-micron resolution maps of topography, mechanical properties, and chemical composition in individual cell walls in plant tissues are possible. This is an excellent technical advancement in the field. The paper is well written and the results are presented in a clear manner. The discussion is also adequate. I suggest some minor corrections that I list below:

1. The abstract should clearly state that this technique cannot be used in living plants, and that it can only be employed in plant sections that have been subject to chemical fixation.

2. Perhaps the most disappointing part of the paper is that the AFM part does not give quantitative results of stiffness. The use of an AFM contact resonance technique is very much appreciated by this reviewer, but I miss an explanation of why the results are just given as phase shift maps. Contact resonance techniques (not using PLL) have proved to be suited to reproducing correct values of stiffness of plant cell walls (Seifert et al. 2021). A good discussion of why quantitative stiffness values are not given here is needed in section 2.1.

3. Adding Figure S1(B) to the figures of the main text would make the paper easier to understand and more attractive. This figure corresponds to the AFM topography that is correlated to the measurements in Figs. 1, 3-5. The paper would become clearer if this AFM topography image is added to Fig.1 and/or Fig. 3.

---

## [Reviewer Report]

*Comments to Author*: The manuscript by Bilkey and co-workers entitled “Correlated mechanochemical maps of Arabidopsis thaliana primary cell walls using atomic force microscope-infrared spectroscopy” reports the results of AFM-IR analysis of Arabidopsis thaliana epidermal cell wall.

The manuscript presents preliminary results of one AFM-IR spectroscopic analysis of the plant tissue. This lowers the scientific rigor and reproducibility of this work. One can expect that 10-20 different areas on several plants have to be investigated and results analyzed to make any conclusions.

Unfortunately, the quality of both AFM and AFM-IR results is also significantly below the level required for any academic paper. By all means, the tissue surface obtained after embedding was still too rough for AFM imaging. Perhaps, the authors could test other AFM tips to improve the quality of plant tissue imaging.

The reviewer is also convinced that deconvolution of the AFM-IR spectra collected from plant tissue by several FTIR spectra of commercially available standards is entirely wrong and misleading. First, any plant tissue contains hundreds of compounds. Secondly, what is the relevance of commercially available soy protein to proteins of Arabidopsis thaliana? The authors should be aware that amide I band is very sensitive to the changes in protein secondary structure. Thus, small changes in the relative amount of alpha-helix vs unordered protein cause significant changes in AFM-IR spectra. Therefore, a spectrum of a ‘random’ commercially available protein becomes entirely useless for such deconvolution. This concerns all biological molecules, including cellulose and xylan.

---

## [Reviewer Report]

*Comments to Author*: Your manuscript entitled “Correlated mechanochemical maps of Arabidopsis thaliana primary cell walls using atomic force microscope-infrared spectroscopy” has been fully evaluated by two independent peer reviewers. The reviews are contrasted. One reviewer appreciated “an excellent technical advancement in the field”, while the other one noted a lack of repetitions and called into question the relevance of the FTIR spectral analysis. Both reviewers were critical of the quality of the AFM measurements.

For the revised version of the manuscript, could you carry out new measurements to strengthen the reliability and reproducibility of your results, and if possible improve their quality and get quantitative results of stiffness? If this is not possible, please provide justification.

In any case, the revised manuscript should better explain the limitations of the technique and the choices made.

---

## [Reviewer Report]

Dear Drs. Hamant and Hartmann,

We greatly appreciate your handling of our manuscript entitled, “Correlated mechanochemical maps of Arabidopsis thaliana primary cell walls using atomic force microscope-infrared spectroscopy” (QPB-22-0008).

We thank the reviewers for their thoughtful and constructive criticism of our work and the Associate Editor for summarizing the main points to address. We have addressed all concerns raised by the reviewers in our Response to Reviewers Comments document and have revised the text accordingly. These changes are in red in the revision.

We believe that our responses to the points raised have made our work stronger and clearer and we hope that this version is now acceptable for publication in Quantitative Plant Biology.

Sincerely,

Ram Dixit

---

## [Reviewer Report]

*Comments to Author*: The paper has demonstrated the value of combining AFM and infrared spectroscopy for quantitative plant biology, and all the initial concerns of this reviewer have been adequately addressed in the revised manuscript.

---

## [Reviewer Report]

*Comments to Author*: The authors addressed all raised concerns and suggestions. I suggest accepting the manuscript in its current from

---

## [Reviewer Report]

*Comments to Author*: The reviewers of the initial version are satisfied with your revised manuscript, which addressed all their concerns.